# Detection of Unresolved Targets for Wideband Monopulse Radar

**DOI:** 10.3390/s19051084

**Published:** 2019-03-03

**Authors:** Tianyi Tsai, Zhiqiang Liao, Zhiquan Ding, Yuan Zhao, Bin Tang

**Affiliations:** 1Research and Development Department, Sichuan Institute of Aerospace Electronics of China, No. 105, Yiduzhong Road, Chengdu 610100, China; liaozhiqiang@126.com (Z.L.); dingzhiquan@126.com (Z.D.); 2School of Information and Communication Engineering, University of Electronic Science and Technology of China, No.2006, Xiyuan Avenue, West Hi-tech Zone, Chengdu 611731, China; zy_uestc@outlook.com (Y.Z.); bint@uestc.edu.cn (B.T.)

**Keywords:** wideband monopulse radar, unresolved targets, Gaussian mixture model

## Abstract

Detecting unresolved targets is very important for radars in their target tracking phase. For wideband radars, the unresolved target detection algorithm should be fast and adaptive to different bandwidths. To meet the requirements, a detection algorithm for wideband monopulse radars is proposed, which can detect unresolved targets for each range profile sampling points. The algorithm introduces the Gaussian mixture model and uses a priori information to achieve high performance while keeping a low computational load, adaptive to different bandwidths. A comparison between the proposed algorithm and the latest unresolved target detection algorithm Joint Multiple Bin Processing Generalized Likelihood Ratio Test (JMBP GLRT) is carried out by simulation. On Rayleigh distributed echoes, the detection probability of the proposed algorithm is at most 0.5456 higher than the JMBP GLRT for different signal-to-noise ratios (SNRs), while the computation time of the proposed algorithm is no more than two 10,000ths of the JMBP GLRT computation time. On bimodal distributed echoes, the detection probability of the proposed algorithm is at most 0.7933 higher than the JMBP GLRT for different angular separations of two unresolved targets, while the computation time of the proposed algorithm is no more than one 10,000th of the JMBP GLRT computation time. To evaluate the performance of the proposed algorithm in a real wideband radar, an experiment on field test measured data was carried out, in which the proposed algorithm was compared with Blair GLRT. The results show that the proposed algorithm produces a higher detection probability and lower false alarm rate, and completes detections on a range profile within 0.22 ms.

## 1. Introduction

For wideband monopulse radars, the unresolved targets may cause error in the target tracking phase. To detect the presence of unresolved targets, researchers [1,2] studied the relationship between the imaginary part of the monopulse ratio and the number of targets unresolved, and revealed that when only a single target exists, the imaginary part is 0, while when there are multiple targets unresolved, the imaginary part is not 0. Based on that fact, a detection algorithm is proposed, but the algorithm is sensitive to noise, and the setting of the detection threshold is dependent on the target’s DOA (Direction of Arrival) and signal-to-jamming ratio. Blair proposed a Generalized Likelihood Ratio Test (Blair GLRT) algorithm for the detection of two unresolved Rayleigh targets that outperforms the former algorithms [3], and LI Chaowei [4] extended Blair’s idea to the detection of three unresolved targets. Besides the Rayleigh assumption, Chaumette [5] and Nikkle [6] derived the mean and variance of the in-phase part of the monopulse ratio, assuming the amplitude of target echoes being a mixture of Rician and Rayleigh distributions [7]. Based on that, Yong Yang [8] proposed an algorithm to detect the presence of chaff centroid jamming, a typical unresolved target, using the in-phase part of the monopulse ratio and the predicted target parameters, which are provided by a tracking algorithm and GPS/INS (Global Positioning System/Inertial Navigation System) data. However, the proposed algorithm was for low resolution radars. Zhang [9] considered a more common case, that the targets were located between range profile sampling points (also referred to as range profile samples or samples hereafter), and developed a maximum likelihood (ML)-based algorithm to estimate angles of multiple unresolved targets. Isaac [10] and Nandakumaran [11] incorporated the detection of unresolved targets into tracking and proposed particle filter-based algorithms to improve detection performance, but the computational load was high, especially when trying to increase the number of particles to achieve higher performance. Glass [12] proposed the JMBP GLRT algorithm based on Zhang’s model to detect unresolved Rayleigh targets. The proposed algorithm has a lower false alarm rate as well as higher detection probability than Blair’s GLRT, but the detection needs to calculate the ML estimation of target parameters, which results in a heavy computational load.

The studies above all assume that the amplitude of target echo is Rayleigh- or Rician-distributed. The assumptions above only hold when there are a large number of scattering centers within resolution bins (e.g., range bin, range-doppler bin, etc.). As radar resolution improves, the number of scattering centers within resolution bins decreases sharply and the above assumptions can no longer hold. Based on field test measured data, Du Lan [13,14] analyzed the statistical characteristics of echo amplitude in each range bin of high-resolution range profiles, and concluded that the statistical distribution of the echo amplitude of a certain range bin depends on the number and type of scattering centers in the bin. When there is only one dominant scattering center, the amplitude is unimodally distributed. When there are a few dominant centers, the amplitude is multimodally distributed, which cannot be modelled by Rayleigh and Rician distributions. The diversity of radar resolution and target structure results in the diversity of echo amplitude statistical distributions, and a practical detection algorithm needs to be adaptive to different radar resolutions. On the other hand, the time and computation resource of radars for detecting unresolved targets are limited; algorithms based on ML estimation may not be the best choice. To achieve high performance while keeping low computational load, we propose a new algorithm named GBD (GMM-based Bayesian Detector) which has the following features:The Gaussian Mixture Model (GMM) is introduced to model the probability density function (PDF) of the echo monopulse ratio. Theoretically, the GMM can fit any form of PDF. Taking the advantage of the GMM, the algorithm can fit the PDF of an echo monopulse ratio by processing measured data instead of assuming. Therefore, the proposed algorithm is more adaptive to radars of different resolution than the existing algorithms.The detection result of the previous detection period is used to improve detection performance. Because the relative movement between targets is continuous, the detection result of the previous detection period can be used as a priori information for the current detection. Inspired by the PDAF–BD (Probabilistic Data Association Filter-Bayesian Detector) algorithm [15], we establish the tracking of the unresolved targets and use the detection results of the previous detection period to derive the a priori information, and incorporate it into a Bayesian detector to detect unresolved targets.

In this paper, Section 2 introduces the modelling of echoes, Section 3 introduces the proposed detection algorithm, and Section 4 tests the performance and verifies the effectiveness of the proposed algorithm by simulation and experimentation on measured data.

## 2. Signal Modelling

In wideband monopulse radars, the matched filtered echoes of unresolved targets are within the same resolution bin and are sampled with period Δt. Assuming there are N targets and the sth target is composed of $N_{s}$ scattering centers, the samples of the in-phase and quadrature part of the matched filtered echoes are [16,17]
(1)$$s_{i}\left(j\right)=\overset{N}{\underset{s=1}{\sum }}\overset{N_{s}}{\underset{l=1}{\sum }}A_{sl}\cos(\phi_{sl})r\left(j\Delta t-\tau_{sl}\right)+n_{si}\left(j\Delta t\right)$$
(2)$$s_{q}\left(j\right)=\overset{N}{\underset{s=1}{\sum }}\overset{N_{s}}{\underset{l=1}{\sum }}A_{sl}\sin(\phi_{sl})r\left(j\Delta t-\tau_{sl}\right)+n_{sq}\left(j\Delta t\right)$$
(3)$$d_{i}\left(j\right)=\overset{N}{\underset{s=1}{\sum }}\overset{N_{s}}{\underset{l=1}{\sum }}A_{sl}\eta_{s}\cos(\phi_{sl})r\left(j\Delta t-\tau_{sl}\right)+n_{di}\left(j\Delta t\right)$$
(4)$$d_{q}\left(j\right)=\overset{N}{\underset{s=1}{\sum }}\overset{N_{s}}{\underset{l=1}{\sum }}A_{sl}\eta_{s}\sin(\phi_{sl})r\left(j\Delta t-\tau_{sl}\right)+n_{dq}\left(j\Delta t\right)$$
where $s_{i}\left(j\right)$, $s_{q}\left(j\right)$ are the jth sample of the in-phase and quadrature parts of the received sum channel signal; $d_{i}\left(j\right)$, $d_{q}\left(j\right)$ are the jth sample of the in-phase and quadrature part of the received azimuth-difference channel signal; $A_{sl}$ is the voltage signal amplitude of the lth scattering center of the sth target; $\phi_{sl}$ is the echo phase of the lth scattering center of the sth target; r(t) is the known matched filter response of the transmitted pulse; $n_{si}$, $n_{sq}$ are sum channel zero-mean Gaussian noise processes; $n_{di}$, $n_{dq}$ are azimuth-difference channel zero-mean Gaussian noise processes; $\tau_{sl}$ is the round trip time delay from the lth scattering center of the sth target; and $\eta_{s}$ is the DOA parameter of the sth target. The monopulse ratio of the jth sample can be calculated by
(5)$$y_{R}\left(j\right)=\left[d_{i}\left(j\right)s_{i}\left(j\right)+d_{q}\left(j\right)s_{q}\left(j\right)\right]/[s_{i}{\left(j\right)}^{2}+s_{q}\left(j{)}^{2}\right]$$
(6)$$y_{I}\left(j\right)=[d_{q}\left(j\right)s_{i}\left(j\right)-d_{i}\left(j\right)s_{q}\left(j\right)]/[s_{i}{\left(j\right)}^{2}+s_{q}\left(j{)}^{2}\right]$$
where $y_{R}\left(j\right)$ and $y_{I}\left(j\right)$ are the in-phase and quadrature part of the monopulse ratio of the jth sample, respectively. For monopulse radar, the unresolved targets cause a larger angular glint G and $y_{I}\left(j\right)$ than resolved targets [1,3], therefore, G and $y_{I}\left(j\right)$ can be used as the features to detect unresolved targets. Other features such as range glint [18], phase change [19], Doppler jitter, and polarization can also be easily added to the feature vector so as to use more information. In a radar processing period in which the radar performs a set of signal and data processing routines including detecting unresolved targets and tracking, a certain number of independent echoes are received, and the features are calculated based on the echoes. The independence of echoes can be achieved by many means, such as frequency hopping or lengthening the time interval between two echoes by experience replay, a technique widely used in reinforce learning [20].

The observed SNR of the jth sample at time k is defined as
(7)$$R_{O}{}^{k}\left(j\right)=Re{\left[S_{k}\left(j\right)\right]}^{2}/\sigma_{\Sigma }^{2}$$
where $S_{k}\left(j\right)$ is the complex signal of the sum channel of the *j*th sample at time k, and $\sigma_{\Sigma }$ is the variance of the sum channel noise. Therefore, for $K_{0}$ echoes, the angular glint of each echo is
(8)$${\hat{G}}_{k-n}\left(j\right)=y_{R}{}^{k-n}\left(j\right)-y_{R}^{\prime }{}^{k}\left(j\right),n=0,1,\cdots ,K_{0}-1$$
where
(9)$$y_{R}^{\prime }{}^{k}\left(j\right)=\overset{K_{0}-1}{\underset{n=0}{\sum }}R_{O}{}^{k-n}\left(j\right)y_{R}{}^{k-n}\left(j\right)/\overset{K_{0}-1}{\underset{n=0}{\sum }}R_{O}{}^{k-n}\left(j\right)$$
because the noise of the sum channel and azimuth-difference channel are random and independent of each other, a lower $R_{O}{}^{k}\left(j\right)$ means more randomness in the estimated glint ${\hat{G}}_{k-n}\left(j\right)$ and the imaginary part of the monopulse ratio $y_{I}{}^{k}\left(j\right)$ of the jth sample at time k. In addition, it is the magnitude of ${\hat{G}}_{k-n}\left(j\right)$ and $y_{I}{}^{k}\left(j\right)$ instead of their value that rises in the presence of unresolved targets [1]. Therefore, it is reasonable to have the magnitude of ${\hat{G}}_{k-n}\left(j\right)$ and $y_{I}{}^{k}\left(j\right)$ weighted by $R_{O}{}^{k}\left(j\right)$ as the feature, which means the feature vector of the jth sample at time k is
(10)$$x_{k}^{j}={\left[x_{k}^{1}\left(j\right),x_{k}^{2}\left(j\right)\right]}^{T}$$
where $x_{k}^{1}\left(j\right)$ and $x_{k}^{2}\left(j\right)$ is calculated by
(11)$$x_{k}^{1}\left(j\right)=GRG^{T}$$
(12)$$x_{k}^{2}\left(j\right)=Y_{I}RY_{I}{}^{T}$$
in which
(13)$$R=diag\left(R_{O}{}^{k-K_{0}+1}\left(j\right),R_{O}{}^{k-K_{0}+2}\left(j\right),\cdots ,R_{O}{}^{k}\left(j\right)\right)$$
(14)$$G=\left[{\hat{G}}_{k-K_{0}+1}\left(j\right),{\hat{G}}_{k-K_{0}+2}\left(j\right),\cdots ,{\hat{G}}_{k}\left(j\right)\right]$$
(15)$$Y_{I}=\left[y_{I}{}^{k-K_{0}+1}\left(j\right),y_{I}{}^{k-K_{0}+2}\left(j\right),\cdots ,y_{I}{}^{k}\left(j\right)\right]$$

In the typical existing algorithms [3,4,5], the analytical form of the joint PDF of $x_{k}^{j}$ is then derived under the assumption of amplitude and phase distributions of echoes. But this idea may not be applicable to wideband radars because, firstly, the joint PDF of $x_{k}^{j}$ needs to be derived for every typical radar resolution and target structure, which means a heavy workload; secondly, there is no guarantee that the analytical form of the joint PDF of $x_{k}^{j}$ exists for every resolution; thirdly, when adding more features to $x_{k}^{j}$, a new joint PDF needs to be derived, but the joint PDF of the new $x_{k}^{j}$ could also have no analytical form. To solve the problem above, we introduce GMM to model the joint PDF of $x_{k}^{j}$, so that the algorithm can adapt to different resolutions by learning from measured data, and keep the analytical form of the joint PDF unchanged for radars of different bandwidth and $x_{k}^{j}$s of different dimensions. The GMM of $x_{k}^{j}$ can be expressed as
(16)$$\{\begin{array}{l} p\left(x_{k}^{j}\right)=\sum_{c=1}^{K_{c}}\omega_{c}\eta \left(x_{k}^{j};\mu_{c},\Sigma_{c}\right)\\ \eta \left(x_{k}^{j};\mu_{c},\Sigma_{c}\right)=\frac{1}{\sqrt{{\left(2\pi \right)}^{n_{d}}\Sigma_{c}}}e^{-\frac{1}{2}{\left(x_{k}^{j}-\mu_{c}\right)}^{T}\Sigma_{c}\left(x_{k}^{j}-\mu_{c}\right)} \end{array}$$
where $p\left(x_{k}^{j}\right)$ is the PDF of $x_{k}^{j}$, $K_{c}$ is the number of Gaussian components of GMM, $\omega_{c}$ is the weight of every components, satisfying $\overset{K_{c}}{\underset{j=1}{\sum }}\omega_{c}=1$; $\eta \left(x_{k}^{j};\mu_{c},\Sigma_{c}\right)$ is the PDF of the cth Gaussian component, $\mu_{c}\;and\;\Sigma_{c}$ are the mean and variance, respectively, and $n_{d}$ is the number of dimensions of $x_{k}^{j}$. The training of GMM can be found in many literatures [21], so we choose not to elaborate on it here.

## 3. Detection of Unresolved Targets

For extended targets, the purpose of the proposed algorithm is to detect the presence of unresolved scattering centers for each range profile sample. Therefore, we define the events as follows:

$H_{0}$: there are no unresolved scattering centers; the signal of the range profile sample is composed of the echo of a single scattering center and noise.

$H_{1}$: there exist unresolved scattering centers; the signal of the range profile sample comprises echoes of multiple scattering centers of unresolved targets and noise.

The Bayesian detector can be written as
(17)$$\frac{p\left(x_{k}^{j}|H_{1}\right)P\left(H_{1}\right)\left(c_{10}-c_{11}\right)}{p\left(x_{k}^{j}|H_{0}\right)P\left(H_{0}\right)\left(c_{01}-c_{00}\right)}\begin{array}{c} \overset{H_{1}}{\geq }\\ \underset{H_{0}}{<} \end{array}1$$
where $p\left(x_{k}^{j}|H_{1}\right)$ and $p\left(x_{k}^{j}|H_{0}\right)$ are the conditional PDF of $x_{k}^{j}$, which is modelled by GMM; $P\left(H_{1}\right)$ and $P\left(H_{0}\right)$ are the a priori probability of event $H_{1}$ and $H_{0}$, respectively, and are calculated according to the detection results of the previous radar processing period; and $c_{ij}$ is the cost when deciding $H_{j}$ while $H_{i}$ is true.

The proposed algorithm treats a group of unresolved scattering centers as a kind of special target named “u-target” in this paper. It is reasonable to believe that a sample closer to the predicted position of a u-target is more likely to be the sample of the u-target. Therefore, we set up tracking of the u-targets and calculate the a priori possibility P(H1) and P(H0) for each sample of the current radar processing period based on the tracking information of the u-targets of the previous radar processing period. The u-target is treated as a single resolved target whose motion model is of the same type as the model of targets that forms the u-target. The model is linear or Gaussian in most scenarios. Therefore, the Kalman filter is appropriate for the tracking of u-targets [22]. To simplify the discussion, we assume that there is only one u-target in the scenario, for multiple u-targets, each of them can be tracked independently. The state model and measurement model of the u-target are
(18)yk|k−1=Fyk−1|k−1+wk−1
(19)zk|k=Hyk|k+vk
where yk|k−1∈ℝn is the state vector of the u-target whose length is n, including the position and radial velocity, the corresponding measurement vector is zk|k∈ℝm, which is the position of the u-target, F and H are the state transition matrix and the measurement matrix respectively, and wk∼N(0,Q) and vk∼N(0,R) are process noise and measurement noise, which are Gaussian-distributed and are independent of each other. Therefore, H and R can be considered known and fixed. During the short time of unresolved target detection, the pattern of relative movements of targets basically remains the same. Therefore, F and Q can be considered as known and fixed. Define ckj as the jth sample at time k (the kth radar processing period). The steps to detect the presence of the u-target is

(1) Calculate ${\hat{y}}_{k|k-1}$ at time k
(20)$${\hat{y}}_{k|k-1}=F{\hat{y}}_{k-1|k-1}+w_{k-1}$$

(2) Calculate the covariance of the estimation error
(21)$$P_{k|k-1}=FP_{k-1|k-1}F^{T}+Q$$

Then the estimation error of the u-target position at time k is
(22)$$S_{k|k-1}=HP_{k|k-1}H^{T}+R$$

(3) Detect the sample of the u-target and data association

Obviously, samples that are closer to the predicted position of the u-target $HF{\hat{y}}_{k-1|k-1}$ are more likely to be the sample of the u-target, which means the samples have higher a priori probability. In order to determine the a priori probability $P_{k}^{j}\left(H_{1}\right)$ of the jth sample, the distance between $c_{k}^{j}$ and $HF{\hat{y}}_{k-1|k-1}$ needs to be calculated first.

(23)$$v_{k}^{j}=c_{k}^{j}-HF{\hat{y}}_{k-1|k-1}$$

The a priori probabilities $P_{k}^{j}\left(H_{1}\right)$ and $P_{k}^{j}\left(H_{0}\right)$ of the jth sample are functions of $v_{k}^{j}$, and the closer the jth sample is to $HF{\hat{y}}_{k-1|k-1}$, the larger the $P_{k}^{j}\left(H_{1}\right)$ is. Therefore, we have $P_{k}^{j}\left(H_{1}\right)\propto {\left|2\pi S_{k|k-1}\right|}^{-1/2}exp(-\frac{1}{2}v_{k}^{j}{}^{T}S_{k|k-1}^{-1}v_{k}^{j})$. Because $P_{k}^{j}\left(H_{1}\right)$ of all samples at time k shares the same term ${\left|2\pi S_{k|k-1}\right|}^{-1/2}$, this term does not contribute to the comparison among a priori probabilities of samples and therefore, the a priori probability that the jth sample is a u-target sample is
(24)$$\{\begin{array}{l} P_{k}^{j}\left(H_{1}\right)\propto e^{-\frac{1}{2}v_{k}^{j}{}^{T}S_{k|k-1}^{-1}v_{k}^{j}}\\ P_{k}^{j}\left(H_{0}\right)=1-P_{k}^{j}\left(H_{1}\right) \end{array}$$

The Bayesian detector can then be rewritten as
(25)$$\frac{p\left(x_{k}^{j}|H_{1}\right)}{p\left(x_{k}^{j}|H_{0}\right)}\frac{\lambda_{c}}{\exp\left(\frac{1}{2}v_{k}^{j}{}^{T}S_{k|k-1}^{-1}v_{k}^{j}\right)-1}\begin{array}{c} \overset{H_{1}}{\geq }\\ \underset{H_{0}}{<} \end{array}1$$
where $x_{k}^{j}$ is the feature vector of the jth bin at time k, and $\lambda_{c}=\lambda \left(c_{10}-c_{11}\right)/\left(c_{01}-c_{00}\right)$, where λ is the weight of $1/\exp\left(\frac{1}{2}v_{k}^{j}{}^{T}S_{k|k-1}^{-1}v_{k}^{j}\right)-1$.

After the test of all the samples of the range profile, a set of samples ${\left\{z_{k}^{c}\right\}}_{c=1}^{n_{c}}$ are detected as samples of the u-target, where $n_{c}$ is the number of u-target samples. The nearest neighbor rule is adopted for data association, and the distance between the measurement and $H{\hat{y}}_{k|k-1}$ is
(26)$$d_{k}^{c}=v_{k}^{c}{}^{T}S_{k|k-1}^{-1}v_{k}^{c}$$
where $v_{k}^{c}=z_{k}^{c}-H{\hat{y}}_{k|k-1}$. We then choose the $z_{k}^{c}$ that yields the smallest $d_{k}^{c}$ as the true measurement for the follow-up steps.

(4) Calculate the Kalman gain

(27)$$K_{k}=P_{k|k-1}H^{T}{\left(HP_{k|k-1}H^{T}+Q\right)}^{-1}$$

(5) Target state estimation
(28)$${\hat{y}}_{k|k}={\hat{y}}_{k|k-1}+K_{k}\left(z_{k}^{c}-H{\hat{x}}_{k|k-1}\right)$$

(6) Update error covariance
(29)$$P_{k|k}=\left(I-K_{k}H\right)P_{k|k-1}$$

If the tracking of the u-target is not established, the a priori probability cannot be calculated, it is reasonable to assume $P_{k}^{j}\left(H_{1}\right)$=$P_{k}^{j}\left(H_{0}\right)$, which means only the GMM is used to detect u-target samples. The form of the detector is as follows with $\lambda_{c}^{\prime }=\left(c_{10}-c_{11}\right)/\left(c_{01}-c_{00}\right)$:(30)$$\lambda_{c}^{’}\frac{p\left(x_{k}^{j}|H_{1}\right)}{p\left(x_{k}^{j}|H_{0}\right)}\begin{array}{c} \overset{H_{1}}{\geq }\\ \underset{H_{0}}{<} \end{array}1$$

The flowchart of GBD algorithm is shown in Figure 1.

The GBD algorithm has two work modes, mode 1 is functional when the tracking of u-targets is not established. In this mode, only the GMM is used to detect samples of u-targets. Mode 2 is functional when the tracking of u-targets is established. In this mode, both GMM and a priori information are used. The establishment and deletion of tracks is designed according to a specific tracking scenario and is performed after the detection of all samples in every radar processing period.

## 4. Comparison and Evaluation

In this part, a series of simulations are carried out to compare GBD with the JMBP GLRT, and then an experiment based on measured data is carried out to evaluate the performance of GBD in a real wideband monopulse radar. 

The simulation is based on a typical scenario shown in Figure 2, in which a u-target is in the fourth bin which is composed of the fourth scattering center of target 1 and the only scattering center of target 2. There is only one scattering center of target 1 in each of the other bins. The targets have no relative movement during the simulations. 

### 4.1. Simulation 1: Performance of GBD and JMBP GLRT on Echoes of Different SNR

In this simulation, the echoes are Rayleigh-distributed. For comparison, the definition of scattering center parameters is the same as those for JMBP GLRT [12], that is, the total SNR of the lth scattering center of the sth target is defined as $R_{t,sl}=N\beta_{sl}^{2}/\sigma_{\Sigma }^{2}$, with N being the number of pulses used in a radar processing period, $\beta_{sl}^{2}$ being the variance of $A_{sl}\cos(\phi_{sl})$ and $A_{sl}\sin(\phi_{sl})$, and $\sigma_{\Sigma }^{2}$ being the variance of the sum channel noise samples. The range $\tau_{sl}$ is normalized to the range resolution, and the angles $\eta_{s}$ are normalized to the 3 dB beam width. The sub-bin location is defined as $c_{sl}=\left(\tau_{sl}-j_{sl}\Delta t\right)/\Delta t$, where $j_{sl}$ is the first range sample with the energy of the lth scattering center of the sth target, and r(t)=1−t/Δt, where 0≤t≤Δt. The target parameters for the fourth bin are $R_{t,14}=16\;\mathrm{dB}$, $c_{14}=0.3$, $\eta_{1}=-0.5$, and the sub-bin position and DOA parameter of target 2 is $c_{2}=0.6$, $\eta_{2}=0.5$. The target parameters for each of the other bins are $R_{t,1l}=16\;\mathrm{dB}$, $c_{1l}=0.3$, and $\eta_{1}=-0.5$, where l={1,2,3,5,6,7}. For GBD, to simplify the simulation, the echoes in a bin are only sampled by the leading sample point, which means the echo in the fourth bin is sampled by the fourth sampling point, and the fifth sampling point samples the echo in the fifth bin. As the echoes in a certain bin are actually sampled by the leading and end sampling points, the assumption above is equivalent to the method that declares the presence of a u-target in the bin when at least one of the two sampling points is detected as a u-target sample. The echoes of each bin are independent of each other. Aside from the test echoes, another 1000 echoes generated under the same parameter set is used to train the GMM, and the number of GMM components is $K_{c}=5$. We set the system noise variance Q=1.73, and the measurement noise variance R=1. $K_{0}=5$, which is equivalent to M=5 for JMBP GLRT, is set. The total scattering center SNR of target 2 $R_{t,2}$ sweeps from 12 dB to 30 dB, for each $R_{t,2}$, 1000 Monte Carlo simulations are performed, and the percentage of correct detection $P_{d}$ of the two algorithms under the same false alarm rate $P_{fa}\leq 0.02$ is shown in Figure 3a. As is shown, both of the $P_{d}$s of the two algorithms rise as $R_{t,2}$ increases, and GBD performs better. For example, when $R_{t,2}=16\;\mathrm{dB}$, the $P_{d}$ of JMBP GLRT is 0.62, and the $P_{d}$ of GMD is 0.88. This is because GBD uses the samples of echoes as well as tracking information while JMBP GLRT uses the samples only. Aided by the tracking information, GBD lowers the threshold for detecting unresolved targets at the sample that is expected to be a u-target sample and raises the threshold at other samples. Thus, more u-target echoes as well as resolved target echoes are correctly detected.

In Figure 3b, the computation time of GBD is between $1.3\times 10^{-3}\;$s and $1.6\times 10^{-3}\;$s, while the computation time of JMBP GLRT is between 7.6 s and 9.6 s under different $R_{t,2}$. This is because the computation for calculating GMM and Kalman filtering is much lower than solving the ML in JMBP GLRT. If the initial search point for JMBP GLRT is not set to the true value of the target parameters, which is more reasonable in real scenarios as there is no knowledge of the true value of target parameters, the computation time of JMBP GLRT would be even longer.

### 4.2. Simulation 2: Performance of GBD and JMBP GLRT on Echoes of Bimodal Distribution

The amplitude of echoes may not remain Rayleigh-distributed in different resolutions, so detection algorithms should be adaptive to echoes of different distributions while keeping low computational load. In this simulation, the performance of GBD and JMBP GLRT on echoes of different distributions are compared.

In this simulation, the resolution of the radar is high, and the echoes of them are bimodally distributed. We model the echoes by a mixture of two Gaussians according to the results in [14]. In the fourth bin, the parameter set for Gaussian component 1 of the fourth scattering center of target 1 is $\mu_{14}^{1}=3.954,\;\sigma_{14}^{1}=0.3954$, and the parameter set for component 2 of the fourth scattering center of target 1 is $\mu_{14}^{2}=7.908,\sigma_{14}^{2}=0.3954$. The range and angle parameters are $c_{14}=0.3$, $\eta_{1}=0$. The parameter set for Gaussian components of target 2 is the same as target 1, and $c_{2}=0.6$, $\eta_{2}$ sweeps from 0.1 to 0.9 to test the performance of the two algorithms in different angular separations between unresolved targets. In other bins, the parameter set for echo generation is the same as target 1 in the fourth bin, and $c_{1l}=0.3$, $\eta_{1}=0$, where l={1,2,3,5,6,7}. Besides the test echoes, another 1000 echoes are generated under the same parameter set and are used to train the GMM. The number of GMM components is $K_{c}=5$. The normalized histogram of the sum channel power of a sample which samples the echoes of only one scattering center is shown in Figure 4, which is clearly bimodally distributed.

For each $\eta_{2}$, 1000 Monte Carlo simulations are performed for the two algorithms with the initial search point for JMBP GLRT being the true values of the scattering points. In this simulation, 6 pulses are used for detecting unresolved targets. On the echoes generated from the same distribution with identical parameters, the $P_{d}$s of the two algorithms under the same false alarm rate $P_{fa}\leq 0.02$ is shown in Figure 5a. As is shown, both of the $P_{d}$ of the two algorithms rise as $\eta_{2}$ increases, and GBD performs better. This is because the greater angular separation means a stronger unresolved target effect, and GBD is less sensitive to the mismatch of the signal model than JMBP GLRT, producing a significantly higher $P_{d}$ than the latter one. The bimodally distributed data yields more spikes in the likelihood function for JMBP GLRT and, therefore, there is a degradation in the performance of JMBP GLRT.

In Figure 5b, the computation time of GBD is between $0.7\times 10^{-3}\;$s and $0.8\times 10^{-3\;}$s, while the computation time of JMBP GLRT is between 7.2 s and 10.0 s under different $\eta_{2}$. This is because there is no ML solving in GBD. As $\eta_{2}$ increases, the computation time of the two algorithms decreases. This is probably because a larger angular separation of the two targets yields a stronger effect of unresolved targets.

### 4.3. Simulation 3: Performance of GBD when there are more than Two Targets Unresolved

The JMBP GLRT is developed assuming only two targets are unresolved, but in real scenarios, there are probably more than two targets unresolved. In this section, we carry out a simulation to test the performance of GBD when there are more than two targets unresolved. For scenario 1, in which two targets are unresolved, target 2 and the fourth scattering center of target 1 is located in the fourth bin, and the parameters are $R_{t,14}=16\;\mathrm{dB}$, $c_{14}=0.3$, $\eta_{1}=-0.5$, $R_{t,2}=16\;\mathrm{dB}$, $c_{2}=0.6$, $\eta_{2}=0.5$. For scenario 2, in which three targets are unresolved, target 2, target 3, and the fourth scattering center of target 1 are located in the fourth bin, and the parameters are $R_{t,14}=16\;\mathrm{dB}$, $c_{14}=0.25$, $\eta_{1}=-0.5$, $R_{t,2}=16\;\mathrm{dB}$, $c_{2}=0.5$, $\eta_{2}=0$, $R_{t,3}=16\;\mathrm{dB}$, $c_{3}=0.75$, $\eta_{3}=0.5$. For scenario 3, in which four targets are unresolved, target 2, target 3, target 4, and the fourth scattering center of target 1 are located in the fourth bin, and the target parameters are $R_{t,14}=16\;\mathrm{dB}$, $c_{14}=0.2$, $\eta_{1}=-0.5$, $R_{t,2}=16\;\mathrm{dB}$, $c_{2}=0.4$, $\eta_{2}=-0.25$, $R_{t,3}=16\;\mathrm{dB}$, $c_{3}=0.6$, $\eta_{3}=0.25$, $R_{t,4}=16\;\mathrm{dB}$, $c_{4}=0.8$, $\eta_{4}=0.5$. In this simulation, five pulses are used for detecting unresolved targets. Besides the test echoes, another 1000 echoes generated under the same parameters set are used to train the GMM, and the number of GMM components is $K_{c}=5$. The mean of $P_{d}$ and $P_{fa}$ of the GBD in each scenario is shown in Table 1. It can be seen in the table that as the number of targets unresolved increases, the $P_{d}$ increases and the $P_{fa}$ decreases, which means the GBD performs better when more targets are unresolved. This is because the targets are of equal SNR and evenly located in range and angle, thus more targets cause stronger unresolved target effect.

As the unresolved target effect is the very factor that affects radar performance, the weaker unresolved target effect, caused by closely spaced unresolved targets for example, may result in the degradation of GBD performance, but its harmful effect to the radar will also degrade. That means the degradation of GBD performance caused by the weaker unresolved target effect may not necessarily degrade the radar tracking performance.

### 4.4. Experiment: Test on Measured Data

In this section, the performance of GBD is evaluated by field test measured data. The data collection experiment is shown in Figure 6. In the experiment, the radar was installed ashore, the ship was at anchor, and a small ship carrying a corner reflector moved from the position shown in the figure across the ship in range dimension. The distance between the corner reflector and the ship in azimuth dimension is approximately 50 m, corresponding to $0.45\theta_{3\;\mathrm{dB}}$. When the reflector and some part of the ship are in the same range, a u-target is formed. The monopulse radar is in X band, and uses a linear frequency modulation (LFM) signal of 50 MHz bandwidth. During the experiment, the radar beam kept pointing at the ship. The range profiles of the ship and corner reflector produced by the radar are shown in Figure 7. It is obvious in Figure 7 that both the corner reflector and the ship occupy multiple range bins, but the JMBP GLRT is developed assuming that targets are within one range bin. Therefore, the assumption of JMBP GLRT cannot be satisfied and the algorithm is not suitable for the experiment. To provide a benchmark for evaluating the performance of GBD, instead of JMBP GLRT, GBD is compared with Blair GLRT [7], a typical algorithm applicable to the experiment and the very algorithm compared with JMBP GLRT in the literature [12]. 

In the experiment, the radar processing period is the coherent processing interval (CPI), which means the u-target detection and target angle estimation is performed in every CPI. The target angle is estimated by a centroid method
(31)$$A_{\mathrm{target}}=\frac{\sum_{j=1}^{c_{k}}R_{O}{}^{k}\left(j\right)y_{R}^{k}\left(j\right)}{\sum_{j=1}^{c_{k}}R_{O}{}^{k}\left(j\right)}$$

In which $c_{k}$ is the number of samples in the range profile of the target at time k. The target angles of each CPI estimated by Equation (31) without using u-target detection algorithm is shown in Figure 8, in which the horizontal axis is the index of CPIs, corresponding to time, and the vertical axis is the $A_{\mathrm{target}}$ of each CPI. In Figure 8, each angle estimate $A_{\mathrm{target}}$ is calculated by all range profile samples of the CPI that the $A_{\mathrm{target}}$ belongs to, which means no u-target samples is detected and removed before the $A_{\mathrm{target}}$ is calculated.

According to the experiment record, the u-target is formed within the time interval of CPI = 7000–9000. It can be seen in Figure 8, that during this interval, the u-target caused a significantly larger angle glint than other CPIs. At time intervals other than CPI = 7000–9000, the ship and the corner reflector are resolvable in range (e.g., Figure 7.), and there are no unresolved targets.

To evaluate the performance of GBD on the measured data, the data above is used as the test data. For GBD, another 1000 ship echoes and 1000 unresolved target echoes sampled in the same scenario are used to train the ship GMM model and the unresolved targets GMM model, respectively, with the number of GMM components $K_{c}=5$. Set $K_{0}=5$, and with $\lambda_{c}=\lambda_{c}^{’}=0.067$ set to suppress the false alarm rate. According to radar parameters, the system noise variance is set Q=6, and the measurement noise variance R=6. The condition for deleting the track is that no unresolved targets are detected in more than five consecutive CPIs. For Blair GLRT, five consecutive range profiles are used for one detection, the false alarm rate is set to $1\times 10^{-3}$ and a fixed threshold is used to detect the range profile. The samples of u-targets are detected by GBD or Blair GLRT and are removed before calculating $A_{\mathrm{target}}$ with Equation (31). The curves of $A_{\mathrm{target}}$ is shown in Figure 9.

It can be clearly seen in the figure that during period CPI = 7000–9000, $A_{\mathrm{target}}$ calculated with GBD has the smallest glint, and $A_{\mathrm{target}}$ calculated with Blair GLRT has a medium-scale glint. As samples of u-targets cause a larger glint than samples of resolved targets, the curves in Figure 9 indicate that GBD correctly detected and removed more samples of unresolved targets than Blair GLRT, and reduced a more harmful effect of unresolved targets than Blair GLRT in angle estimation. To quantitatively evaluate the effect of GBD and Blair GLRT in radar angle estimation, we calculated the variance of angle estimations in typical time intervals shown in Table 2.

It can be seen from the table that in the time interval CPI = 7000–9000 where the u-target exists, the $A_{\mathrm{target}}$ calculated with GBD has the smallest variance, indicating that the GBD algorithm correctly removed more samples of large angular glint than Blair GLRT and therefore effectively suppressed the angular glint caused by the u-target. In time intervals CPI = 1–6000 and CPI = 10000–13000, targets are resolvable, which means samples being detected as u-target samples in these intervals are false alarms. In the two time intervals, compared with Blair GLRT, the variance of $A_{\mathrm{target}}$ calculated with GBD is closer to the variance of $A_{\mathrm{target}}$ calculated without u-target detection, indicating that GBD removed less samples of large glint than Blair GLRT, meaning that GBD has a lower false alarm rate than Blair GLRT. As one of the main effects of a u-target is increasing angular glint, and the GBD algorithm effectively reduced the effect, we can safely conclude that the GBD algorithm correctly detected samples of u-targets and effectively improved radar angle estimation performance in the presence of u-targets.

In the experiment, the average computation time of GBD in one CPI is $21\times 10^{-4}\;$s, while the average computation time of Blair GLRT in one CPI is $4\times 10^{-4}\;$s, so Blair GLRT is faster than GBD by five times. The two algorithms were programmed by MATLAB in a PC with Intel Core i5-8250 CPU @ 1.8 GHz, 8 GB memory, and a Windows 10 operation system.

Blair GLRT is a typical algorithm applicable to wideband radar. The experiment compared GBD with Blair GLRT on the basis of an LFM radar. As the signal models of the two algorithms are baseband signal models, the modulation of the radar signal has no major influence on the difference in performance of the two algorithms. If the two algorithms are applied to pseudo-noise sequence phase-coded radars, GBD will also have a higher detection probability and lower false alarm rate than Blair GLRT because GBD uses monopulse ratio and tracking information, while Blair GLRT only uses monopulse ratio. GBD also avoids the risk of model mismatch which Blair GLRT suffers in wideband radars. But the computation load of GBD is higher than Blair GLRT because the tracking function and GMM likelihood calculation of GBD takes more computation than the Neyman–Person test of Blair GLRT. The experiment results show that the MATLAB version of GBD takes approximately 0.2 ms to complete detections on a range profile, a time length already comparable to data acquisition periods. In our coding experience, a C language version will run much faster than a MATLAB version, and therefore, the computation time of GBD can be further reduced, meaning GBD has the potential to meet the computation time requirement of real-time processing.

## 5. Conclusions

In this paper, we proposed an algorithm named GBD that detects unresolved targets for wideband monopulse radars. The proposed algorithm models target echo by GMM to be adaptive to different radar bandwidths and use a priori information to achieve high performance while keeping low computational load. Simulation results show that GBD has better detection performance and lower computational load than JMBP GLRT, and is adaptive to echoes of different distributions. The experiment on measured data proved that GBD can correctly detect samples of u-targets and performs better than Blair GLRT in a real wideband monopulse radar. The GBD algorithm can be applied to wideband monopulse radars of different resolutions, and has the potential to meet the computation time requirement of real-time processing. 

For wideband radars, a practical unresolved target detection algorithm should be adaptive to different bandwidths while keeping a low computational load, and this paper introduced our work to meet those requirements. The next goal is to further explore the idea of this paper by optimizing the signal model and tracking algorithm to handle more complicated real scenarios.

## Figures and Tables

**Figure 1 sensors-19-01084-f001:**
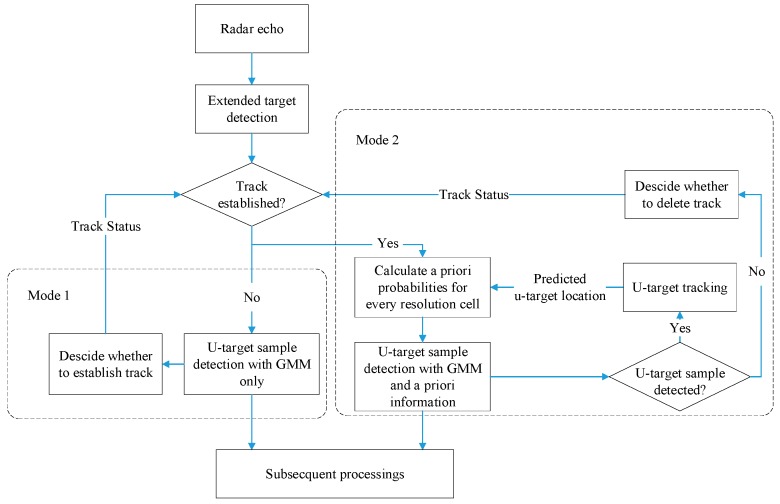
Flowchart of the Gaussian mixture model-based Bayesian detector (GBD).

**Figure 2 sensors-19-01084-f002:**
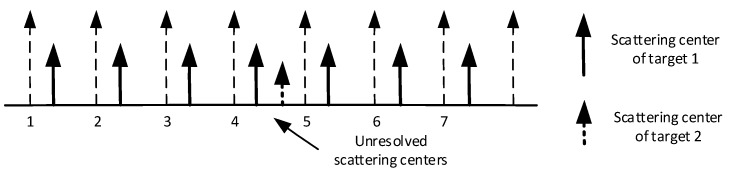
Position of scattering centers.

**Figure 3 sensors-19-01084-f003:**
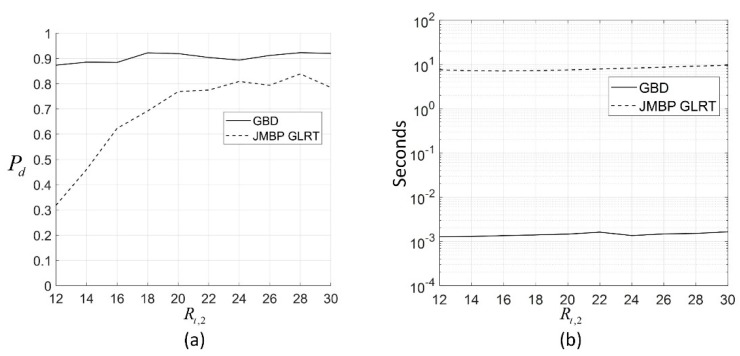
(**a**) $P_{d}$ of the two algorithms for different $R_{t,2}$ with $R_{t,14}=16\;\mathrm{dB}$, $\;c_{14}=0.3$, $\eta_{1}=-0.5$, and $R_{t,1l}=16\;\mathrm{dB}$, $c_{1l}=0.3$, $\eta_{1}=-0.5$, where the $P_{d}$ of the two algorithms are calculated under the same false alarm rate $P_{fa}\leq 0.02$; (**b**) average computation time of the two algorithms for each $R_{t,2}$ with the same parameters set as (**a**); the algorithms are programmed by MATLAB on a PC with Intel Core i5-6300 CPU @ 2.4 GHz, 4 GB memory, and a Windows 10 operation system; interior-point algorithm is used to solve the maximum likelihood (ML) estimation in JMBP GLRT, with the initial search point being the true values of the target parameters.

**Figure 4 sensors-19-01084-f004:**
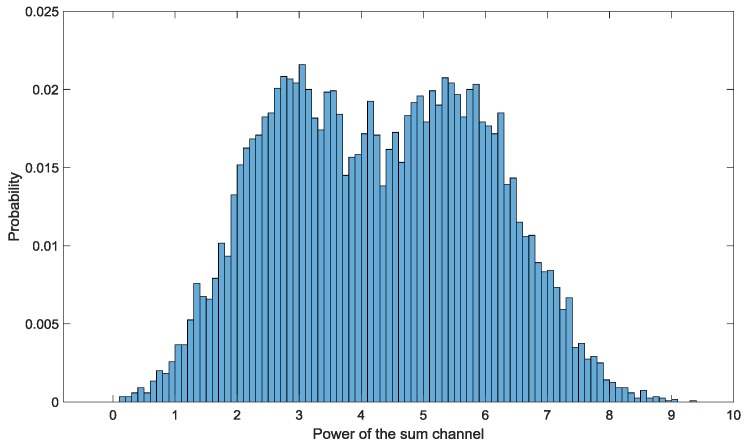
Normalized histogram of the sum channel power of a sample which samples echoes of only one scattering center. The echo of the scattering center is generated by a bimodal distribution composed of two Gaussian components. The parameters set for Gaussian component 1 is $\mu^{1}=3.954,\sigma^{1}=0.3954$ and the parameter set for component 2 is $\mu^{2}=7.908,\sigma^{2}=0.3954$.

**Figure 5 sensors-19-01084-f005:**
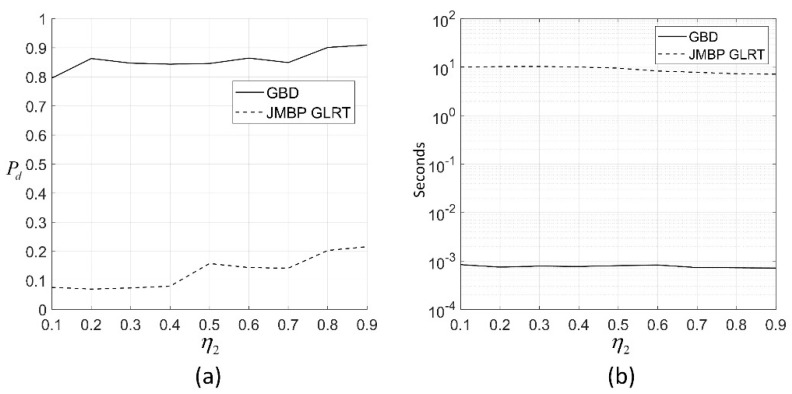
(**a**) $P_{d}$ of the two algorithms for different $\eta_{2}$, with the Gaussian component 1 of the fourth scattering center of target 1 being $\mu_{14}^{1}=3.954,\sigma_{14}^{1}=0.3954$, and the parameter set for component 2 being $\mu_{14}^{2}=7.908,\sigma_{14}^{2}=0.3954$. The range and angle parameters are $c_{14}=0.3$, $\eta_{1}=0$, the parameter set for Gaussian components of target 2 is the same as target 1, and $c_{2}=0.6$, $\eta_{2}$ sweeps from 0.1 to 0.9. The $P_{d}$ of the two algorithms is calculated under the same false alarm rate $P_{fa}\leq 0.02$; (**b**) average computation time of the two algorithms for each $\eta_{2}$ with the same parameters set as (a); the PC platform and ML solver is the same as simulation 1.

**Figure 6 sensors-19-01084-f006:**
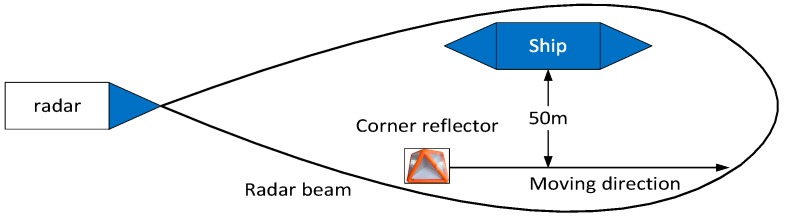
Scenario of data acquisition experiment.

**Figure 7 sensors-19-01084-f007:**
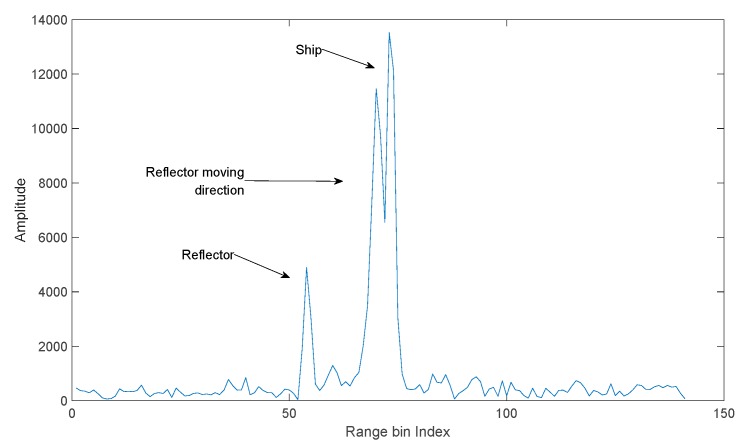
Range profiles of targets.

**Figure 8 sensors-19-01084-f008:**
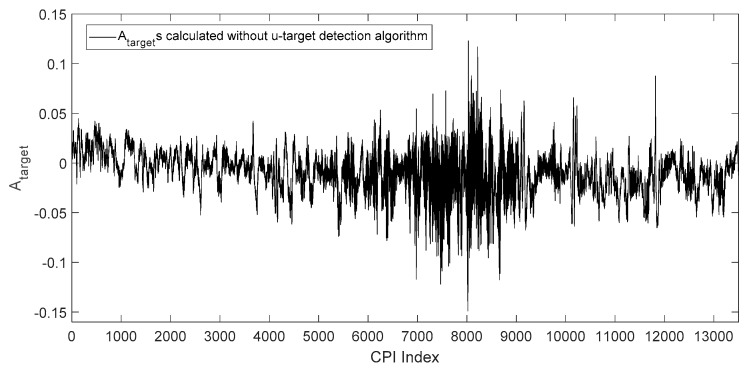
$A_{\mathrm{target}}$s calculated by all range profile samples in each coherent processing interval (CPI).

**Figure 9 sensors-19-01084-f009:**
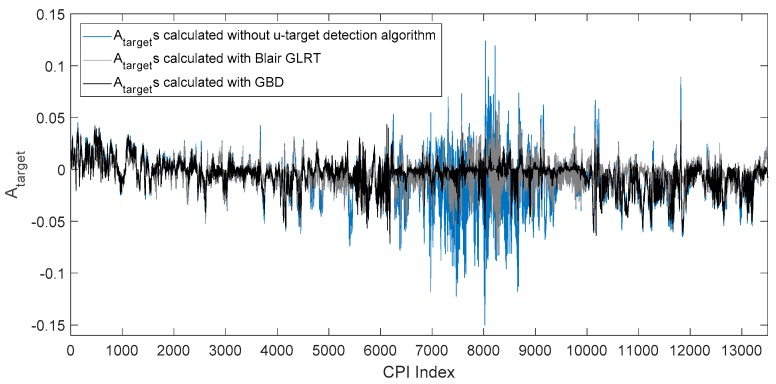
$A_{\mathrm{target}}$s calculated with or without unresolved targets detection algorithms.

**Table 1 sensors-19-01084-t001:** $P_{d}$ and $P_{fa}$ of GBD in each scenario.

Scenario	$P_{d}$	$P_{fa}$
1	0.973	0.004
2	0.993	0.002
3	0.996	0.001

**Table 2 sensors-19-01084-t002:** Variance of target angle in typical time intervals.

CPI Index (Typical Time Intervals)	1–6000	7000–9000	10000–13,000
Variance of $A_{\mathrm{target}}$ calculated without u-target detection	0.30 × 10^−3^	1 × 10^−3^	0.35 × 10^−3^
Variance of $A_{\mathrm{target}}$ calculated with Blair GLRT	0.12 × 10^−3^	0.23 × 10^−3^	0.09 × 10^−3^
Variance of $A_{\mathrm{target}}$ calculated with GBD	0.17 × 10^−3^	0.07 × 10^−3^	0.22 × 10^−3^

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
