# Peer review of "Detection of Unresolved Targets for Wideband Monopulse Radar"

_sensors, 2019, doi:10.3390/s19051084_

Reviewer 1 Report

Please, compare the results of your approach to wideband radar techniques based on the use of pseudonoise sequences (Golay, Barker, PRBS, Costas...). Include advantages and disadvantages.

Author Response

Dear review,

We would like to express our sincere appreciation for your careful reading and invaluable comments to improve this paper. We have carefully considered your advice and modified the paper. The amendments made are mentioned below with reference to appropriate lines of the revised manuscript.

[Comment] Please, compare the results of your approach to wideband radar techniques based on the use of pseudonoise sequences (Golay, Barker, PRBS, Costas...). Include advantages and disadvantages.

[Answer] As JMBP GLRT is not applicable to the experiment, we choose Blair GLRT as a benchmark to evaluate GBD, because Blair GLRT is a typical algorithm applicable to the experiment and the very algorithm compared with JMBP GLRT in the literature. As the signal model of GBD and Blair GLRT are baseband signal models, the modulation of radar signal has no major influence on the performance difference of the two algorithms. We carried out a comparison between GBD and Blair GLRT in the experiment which is based on an LFM radar, and on the basis the experiment results, analyzed the advantages and disadvantages of GBD if the two algorithms are applied to pseudo noise sequences phase-coded radars. The advantage of GBD is that GBD avoids the risk of model mismatch which Blair GLRT suffers in wideband radars and GBD uses more information than Blair GLRT, therefore GBD performs better and are more adaptive. The disadvantage of GBD is that GBD needs more computation than Blair GLRT, but GBD has the potential to meet the requirement of real-time processing. Please refer to Line 425-438 for details.

The PDF version of the revised manuscript is attached to this email.

Thank you again for your comments and we look forward to hearing form you. We would be glad to respond to any further question and comments that you may have.

Yours sincerely,

Tianyi Tsai

Reviewer 2 Report

Major comments:

1.     It would be nice to discuss the accordance between experimental results and simulation results. However, as mentioned in the first sentence of 4.4, JMBP GLRT is not applicable in the experiment. This would be somewhat limited as there’s no comparison performed in the experiment, whereas simulation results have clearly compared the two algorithms in several aspects.

2.     Please be more specific on the results of simulation, and actual experiment. E.g. how much higher in performance and by what standard is the new algorithm? How much lower a computational load is saved, and how adaptive? Please list quantitative results in the abstract so that reader can have a preview snapshot of your achievements.

Minor comments:

3.     Please spell out JMBP GLRT algorithm since it first appears in the abstract ahead of main context.

4.     Table I should have units associated with each parameter

5.     Page 4, Line 179, is it a typo in the subtitle just before “Calculate”?

6.     Page 8, Line 274, where is the table as mentioned in this line?

7.     Please consider black and white printout situation when readers might get confused about which line is which for some figures (Fig. 6&7). A solid line and a dash line would help increase the contrast.

8.     Please revise title of Fig. 7 to reflect the key parameters in the simulation and results. Fig. 6 has set a good example on titling.

Author Response

Dear review,

We would like to express our sincere appreciation for your careful reading and invaluable comments to improve this paper. We have carefully considered your advice and modified the paper. The amendments made are mentioned below with reference to appropriate lines of the revised manuscript.

[Comment 1] It would be nice to discuss the accordance between experimental results and simulation results. However, as mentioned in the first sentence of 4.4, JMBP GLRT is not applicable in the experiment. This would be somewhat limited as there’s no comparison performed in the experiment, whereas simulation results have clearly compared the two algorithms in several aspects.

[Answer] As JMBP GLRT is not applicable in the experiment, we choose Blair GLRT as a benchmark to evaluate GBD, because Blair GLRT is a typical algorithm applicable to the experiment and the very algorithm compared with JMBP GLRT in the literature. The results show that GBD has higher detection probability and lower false alarm rate than Blair GLRT, but Blair GLRT is faster than GBD by 5 times. Although GBD is slower than Blair GLRT, it only takes 0.2ms to complete detections on a range profile, a time length already comparable to data acquisition periods, if GBD is programmed by C language, it has the potential to meet the computation time requirement of real-time processing. The figures (Figure8, Figure9) and Table 2 are updated by new results, and the corresponding text are updated accordingly. (Please refer to Line 367-369, 395-438 for details)

We haven’t found the way to clearly illustrate the three curves in Figure 9 with black and gray lines, so we adopted a colored way. We will contact the editors to solve this problem.

Because calculating the value of detection probability and false alarm rate needs the exact position of the targets (the precision of positioning needs to be higher than 3 meters), but the GPS used in the experiment cannot achieve the required positioning precision, so we cannot directly calculate the detection probability and false alarm rate of GBD and Blair GLRT. As the performance difference of the two algorithms can be seen by analyzing the estimation of , we indirectly compared the two algorithms by analyzing the estimations of .

[Comment 2] Please be more specific on the results of simulation, and actual experiment. E.g. how much higher in performance and by what standard is the new algorithm? How much lower a computational load is saved, and how adaptive? Please list quantitative results in the abstract so that reader can have a preview snapshot of your achievements.

[Answer] We have listed quantitative results of simulations and experiment in the abstract, including detection probability and computation time. (Please refer to Line 17-29 for details)

[Comment 3] Please spell out JMBP GLRT algorithm since it first appears in the abstract ahead of main context.

[Answer] The full name of JMBP GLRT is added to the abstract, other acronyms such as SNR, DOA is also spelled out when they first appear. (Please refer to Line 18-19, 21, 39 for details)

[Comment 4] Table I should have units associated with each parameter

[Answer] Because the parameter  is the probability of detection and  is the probability of false alarm in this paper, there is no unit for them there. (Please refer to Line 349 for details)

[Comment 5] Page 4, Line 179, is it a typo in the subtitle just before “Calculate”?

[Answer] Sorry, it is a typo. The typo has been removed. (Please refer to Line 186 for details)

[Comment 6] Page 8, Line 274, where is the table as mentioned in this line?

[Answer] Sorry, the word ‘table’ actually means Figure 3, the typo has been corrected, (Please refer to Line 280 for details).

.

[Comment 7] Please consider black and white printout situation when readers might get confused about which line is which for some figures (Fig. 6&7). A solid line and a dash line would help increase the contrast.

[Answer] Fig. 3&4 and Fig. 6&7 are grouped respectively to be more compact. The line style of Fig. 3&4 and Fig. 6&7 is adjusted to be clearer, the fonts in these figures are also unified. (Please refer to Line 272-279, 318-325 for details)

[Comment 8] Please revise title of Fig. 7 to reflect the key parameters in the simulation and results. Fig. 6 has set a good example on titling.

[Answer]The titles of Fig.3, Fig.4, Fig.5 are revised to include key parameters. (Please refer to Line 272-279, 303-307, 318-325 for details)

The PDF version of the revised manuscript is attached to this email for reference.

Thank you again for your comments and we look forward to hearing from you. We would be glad to respond to any further question and comments that you may have.

Yours sincerely,

Tianyi Tsai
